# Transcription-factor-dependent enhancer transcription defines a gene regulatory network for cardiac rhythm

Xinan H Yang[1,2,3†], Rangarajan D Nadadur[1,2,3†], Catharina RE Hilvering[4], Valerio Bianchi[4], Michael Werner[5,6], Stefan R Mazurek[1,2,3], Margaret Gadek[1,2,3], Kaitlyn M Shen[1,2,3], Joseph Aaron Goldman[7,8], Leonid Tyan[1,2,3], Jenna Bekeny[1,2,3], Johnathon M Hall[5,6], Nutishia Lee[7], Carlos Perez-Cervantes[1,2,3], Ozanna Burnicka-Turek[1,2,3], Kenneth D Poss[7,8], Christopher R Weber[1,2,3], Wouter de Laat[4], Alexander J Ruthenburg[5,6], Ivan P Moskowitz[1,2,3]*

[1]Department of Pediatrics, The University of Chicago, Chicago, United States; [2]Department of Pathology, The University of Chicago, Chicago, United States; [3]Department of Human Genetics, The University of Chicago, Chicago, United States; [4]Hubrecht Institute-Koninklijke Nederlandse Akademie van Wetenschappen, University Medical Center Utrecht, Uppsalalaan, Netherlands; [5]Department of Biochemistry and Molecular Biology, The University of Chicago, Chicago, United States; [6]Department of Molecular Genetics and Cell Biology, The University of Chicago, Chicago, United States; [7]Department of Cell Biology, Duke University School of Medicine, Durham, United States; [8]Regeneration Next, Duke University, Durham, United States

*For correspondence:
imoskowitz@uchicago.edu

[†]These authors contributed equally to this work

Competing interests: The authors declare that no competing interests exist.

**Abstract** The noncoding genome is pervasively transcribed. Noncoding RNAs (ncRNAs) generated from enhancers have been proposed as a general facet of enhancer function and some have been shown to be required for enhancer activity. Here we examine the transcription-factor-(TF)-dependence of ncRNA expression to define enhancers and enhancer-associated ncRNAs that are involved in a TF-dependent regulatory network. TBX5, a cardiac TF, regulates a network of cardiac channel genes to maintain cardiac rhythm. We deep sequenced wildtype and *Tbx5*-mutant mouse atria, identifying ~2600 novel *Tbx5*-dependent ncRNAs. Tbx5-dependent ncRNAs were enriched for tissue-specific marks of active enhancers genome-wide. Tbx5-dependent ncRNAs emanated from regions that are enriched for TBX5-binding and that demonstrated Tbx5-dependent enhancer activity. *Tbx5*-dependent ncRNA transcription provided a quantitative metric of *Tbx5*-dependent enhancer activity, correlating with target gene expression. We identified *RACER*, a novel *Tbx5*-dependent long noncoding RNA (lncRNA) required for the expression of the calcium-handling gene *Ryr2*. We illustrate that TF-dependent enhancer transcription can illuminate components of TF-dependent gene regulatory networks.
DOI: https://doi.org/10.7554/eLife.31683.001

## Introduction

Cardiac rhythm is under tight transcriptional control (e.g. *Nadadur et al., 2016*). Mendelian and genome-wide association genetic studies have implicated cardiac ion channels and transcription factors (TFs) in cardiac conduction and arrhythmias (*Tucker and Ellinor, 2014*). These genes form components of an implied cardiac gene regulatory network, in which TFs drive downstream effectors to maintain normal electrical conduction. Molecular understanding of cardiac rhythm control requires

analysis of the essential TF-driven enhancers that modulate the expression of channel genes, and these regulatory elements are not yet described. TBX5 is a central component of cardiac gene regulatory networks and is required for normal cardiac rhythm (*Nadadur et al., 2016*). For example, we previously demonstrated that TBX5 modulates atrial gene expression to prevent atrial fibrillation (*Nadadur et al., 2016*). In the work described in this paper, we sought to define the essential components of the TBX5-driven gene regulatory network for atrial rhythm.

We attempted to identify TBX5-dependent enhancers by combining datasets for tissue-specific enhancer markers and TBX5 chromatin localization (TBX5 ChIP-seq) in cardiomyocytes (*He et al., 2011*). Previously published datasets describing regions with accessible chromatin defined by DNase I hypersensitivity (*Stergachis et al., 2014*) and the genomic enhancer marker Histone 3 Lysine 27 Acetylation (H3K27Ac) (*Stamatoyannopoulos et al., 2012*) from the whole adult mouse heart were intersected with TBX5 occupancy in the atrial cell line HL-1 (*Claycomb et al., 1998*) (*Figure 1A*). Over 15,000 common genomic locations were identified. We also performed ATAC-seq in HL-1 cells, and intersection with this more limited dataset (with 16,000 accessible regions) identified over 7500 common genomic locations. The high concordance between these datasets suggested a lack of specificity for functional TBX5-dependent enhancers, consistent with observations that genomic enhancer marks and TF ChIP lack specificity or quantitative resolution of enhancer strength (*Zentner and Scacheri, 2012*) and that only a small percentage of these candidate TF-dependent enhancers will prove to be functional *in-vivo* (*Visel et al., 2007*; *Cusanovich et al., 2014*; *Danko et al., 2015*).

The noncoding genome is pervasively transcribed, and a fraction of ncRNAs emanate from enhancers. The relationship between enhancer transcription, enhancer function, and target gene expression has become the subject of vigorous investigation. Recent work indicates that enhancer transcription may be a functional requirement of active enhancers (*Danko et al., 2015*; *Lam et al., 2014*; *Li et al., 2013, 2016*). The level of eRNA expression at enhancers positively correlates with the level of mRNA synthesis at nearby genes, suggesting that eRNA synthesis may occur specifically at enhancers that are actively engaged in promoting mRNA synthesis (*Kim et al., 2010*). Activation of enhancer networks has been linked to the increased transcription of enhancers, suggesting that context-dependent changes in enhancer function may be accompanied by concomitant changes in enhancer transcription (*Wang et al., 2011*). Tissue-specific eRNA expression is a feature of some validated in-vivo enhancers, and eRNA expression signatures can be used to predict tissue-specific enhancers independently of known epigenomic enhancer marks (*Wu et al., 2014*). Separate from the functionality of enhancer-borne RNAs, these and other observations indicate that enhancer and target gene transcription are correlated and that differential enhancer transcription may provide a useful approach for the identification of context-dependent enhancers.

## Results

### *Tbx5*-dependent ncRNA profiling

We hypothesized that tissue-specific TF-dependent enhancers could be identified by defining TF-dependent ncRNA enhancer transcripts. To identify TBX5-dependent noncoding RNAs (ncRNAs), we sequenced non-polyadenylated RNA from the atrium of control ($R26^{CreERt2}$; n = 3) and *Tbx5* mutant ($Tbx5^{fl/fl};R26^{CreERt2}$; n = 4) adult mice, as enhancer-associated RNAs are canonically under polyadenylated (*Kim et al., 2010*). Approximately 40.5 k noncoding transcripts were identified by de novo transcript assembly (*Figure 1B*). Of these, 3610 noncoding transcripts were *Tbx5*-dependent intergenic ncRNAs (FDR < 0.05, FC > 1.5, located at least 1 kbp away from known coding-gene transcription start sites (TSSs) (ENCODE mm10; *Figure 1B*) (*Rosenbloom et al., 2015*). *Tbx5*-dependent ncRNA expression was robust across biological replicates, distinguishing $Tbx5^{fl/fl};R26^{CreERt2}$ samples from $R26^{CreERt2}$ controls (*Figure 1C*). 3067 (85%) *Tbx5*-dependent ncRNA transcripts were located within 2 Mbp upstream or downstream of 2383 *Tbx5*-dependent protein-coding genes, as defined by the RNA-seq of polyA transcripts from the same samples, that is, within the known distance parameters of enhancer–gene-target pairs (*Figure 1D*) (*Nadadur et al., 2016*).

We analyzed the genomic features of *Tbx5*-dependent ncRNAs (*Figure 1B*, *Figure 1—figure supplement 1A*). The vast majority were de novo ncRNAs (>85% were previously unannotated transcripts, based on intersection with the mouse transcriptome [GENECODE, mm10]), with a small

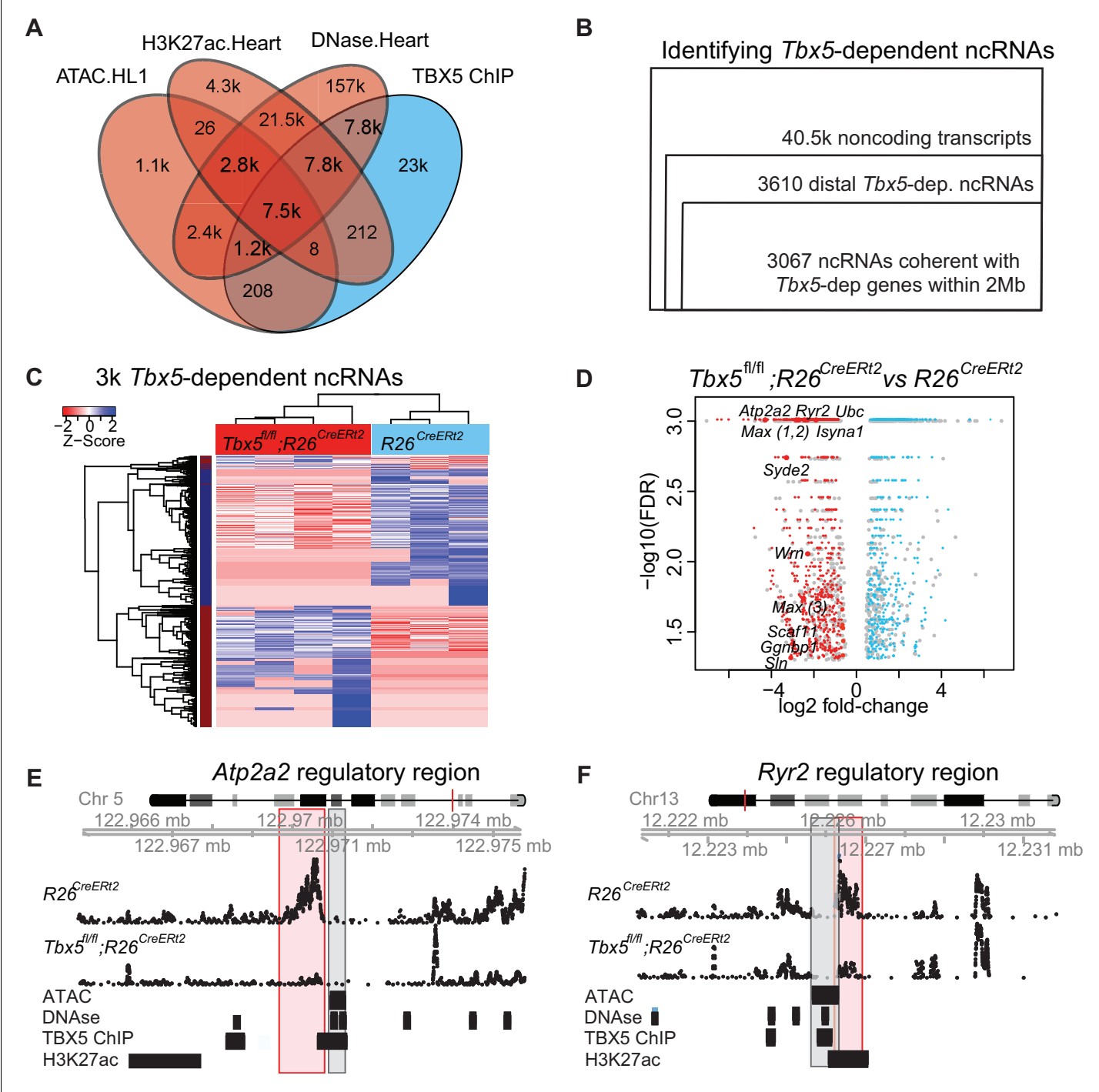

**Figure 1.** TF-dependent noncoding transcription defines regulatory elements. (**A**) Venn-diagram of peak call overlaps for HL-1 ATAC-sequencing (ATAC.HL1), histone 3 lysine 27 acetylation in mouse heart (H3K27ac.Heart, GSE52123), DNase hypersensitivity in mouse heart (DNase.heart, ENCODE) and TBX5 ChIP-seq (TBX5.HL1, GSE21529). (**B**) Workflow for identifying TF-dependent ncRNAs. Total noncoding transcripts from mouse left atrium were narrowed to *Tbx5*-dependent distal intergenic ncRNAs, defined as >1 kbp away from the known transcriptional start sites of known coding genes (GENCODE mm10), and narrowed again to coherent changes with nearby *Tbx5*-dependent genes within a 2-Mbp window. (**C**). Heatmap of identified *Tbx5*-dependent ncRNAs in *Tbx5*$^{fl/fl}$;*R26*$^{CreERt2}$ (red, left) and corresponding *R26*$^{CreERt2}$ control (blue, right) in left atrial tissue. The hierarchical cluster analysis is based on the Euclidean distances of normalized sequencing counts. 1577 *Tbx5*-activated ncRNAs were downregulated after *Tbx5* deletion and 1490 *Tbx5*-repressed ncRNAs were upregulated after *Tbx5* deletion across n = 5 and n = 3 resp. (**D**) Volcano-plot of significantly misregulated TF-dependent ncRNAs, select identifications were labeled by nearest TBX5-dependent gene. Plot of log$_2$ fold-change of ncRNAs in *Tbx5*$^{fl/fl}$;*R26*$^{CreERt2}$ compared to *R26*$^{CreERt2}$ vs –log$_{10}$ false discovery rate (FDR) for the same comparison (FDR < 0.05, |FC| > 2). The ncRNAs within 2 Mb of coherently mis-

*Figure 1 continued on next page*

*Figure 1 continued*

expressed TBX5-dependent genes are red or blue for activated and repressed, respectively. Gray dots represent those ncRNAs without coherently mis-expressed coding genes in the 2-Mb window. (E, F) Example genomic views of two of the most significantly TF-dependent ncRNAs, adjacent to the *Atp2a2* (E) and *Ryr2* (F) genes, respectively. Top track is chromosomal location, followed by the ncRNA read density from $R26^{CreErt2}$ control and $Tbx5^{fl/fl}$; $R26^{CreERt2}$. Below is ATAC-Seq peak call in HL-1 cells, cardiac DNase hypersensitivity (ENCODE), TBX5 ChIP-seq (GSE21529) and cardiac H3k27 acetylation (GSE52123). The identified differential ncRNA is marked in the red box, and the putative regulatory element, as defined by the enhancer marks above, is marked in the gray box.

DOI: https://doi.org/10.7554/eLife.31683.002

The following figure supplements are available for figure 1:

**Figure supplement 1.** Genomic features of the identified TF-dependent ncRNAs recapitulate known features of annotated lincRNAs.

DOI: https://doi.org/10.7554/eLife.31683.003

**Figure supplement 2.** Genomic view of nine TF-dependent ncRNAs (mm9).

DOI: https://doi.org/10.7554/eLife.31683.004

**Figure supplement 3.** Identifying TF-dependent ncRNA targets from TF-dependent expressed genes.

DOI: https://doi.org/10.7554/eLife.31683.005

**Figure supplement 4.** Identifying TF-dependent ncRNA targets from open chromatin regions.

DOI: https://doi.org/10.7554/eLife.31683.006

number previously annotated as long ncRNAs (lncRNAs, 5%), antisense transcripts (2%), or other predicted noncoding transcripts. This observation is consistent with the observation that ncRNAs are highly tissue-specific and that global ncRNA discovery has not been previously reported in the atrium of the heart (*Figure 1—figure supplement 1A*) (*Quinn and Chang, 2016*). *Tbx5*-dependent ncRNAs showed a bimodal length-distribution, with relative enrichments at ~500 bp and at ~2000 bp (*Figure 1—figure supplement 1B*), similar to the distribution of GENCODE-annotated lncRNAs (*Figure 1—figure supplement 1B*). The 40.5 k ncRNA background and Tbx5-dependent ncRNAs both show a preponderance of unidirectional (transcribed from one DNA strand) versus bidirectional transcripts (*Figure 1—figure supplement 1D*). Sequence conservation indicated average Phast-con30 scores of between 0 and 0.2 for the majority of *Tbx5*-dependent ncRNAs, with a distribution similar to those of known lncRNAs and lower than those of promoters (*Figure 1—figure supplement 1D*).

## *Tbx5*-dependent ncRNAs define *Tbx5*-dependent enhancers

We hypothesized that *Tbx5*-dependent ncRNAs marked *Tbx5*-dependent enhancers. *Tbx5*-dependent ncRNAs were generated from locations enriched for enhancer marks genome-wide (*Figure 1E–F*, *Figure 1—figure supplement 2A*). Specifically, we observed enrichment between the 3067 *Tbx5*-dependent ncRNAs and (1) DNAse hypersensitivity in the adult mouse heart, (2) open chromatin (ATAC-sequencing) in HL1 cardiomyocytes, and (3) H3K27Ac in the adult mouse heart (*Figure 1E–F*, *Figure 1—figure supplement 2A*) (*He et al., 2011, 2014*; *Wamstad et al., 2012*). Furthermore, *Tbx5*-dependent ncRNAs defined locations that are enriched genome-wide for TBX5 occupancy in HL-1 cells (*He et al., 2011*) (*Figure 2A*). We further hypothesized that the intersection between *Tbx5*-dependent ncRNAs and regions of open chromatin, a global feature of enhancers (*Andersson et al., 2014*), would mark *Tbx5*-dependent enhancers. We overlapped differential ncRNA expression with cardiac DNAse hypersensitivity (*Stamatoyannopoulos et al., 2012*) and defined chromatin-accessible *Tbx5*-dependent noncoding transcripts as those within a 1-kbp window from DNAse peak-centers. 1703 candidate regulatory elements, from a set of 28,393 TBX5-bound DNAse regions, were identified by this intersection (*Figure 2B*).

We functionally interrogated candidate regulatory elements identified by *Tbx5*-dependent ncRNA transcription for enhancer activity. On the basis of our previous work on the role of TBX5 in the adult atrium (*Nadadur et al., 2016*), we selected a number of regulatory elements near genes that are critical for cardiac physiology and that harbor at least one canonical TBX5-binding motif. These included candidates near the TBX5-dependent calcium-handling genes *Atp2a2*, *Ryr2*, and *Sln*. 11 of the 12 candidate enhancers demonstrated robust *cis*-activity in HL-1 cardiomyocytes, indicative of cardiac enhancer function (P-values from 3.3e-6 to 0.03 [t-test, corrected for multiple comparisons], *Figure 2C*). Overall, these enhancers demonstrated unusually strong *cis*-activation, with nine driving reporter expression more than 10-fold and three more than 50-fold. These enhancers were

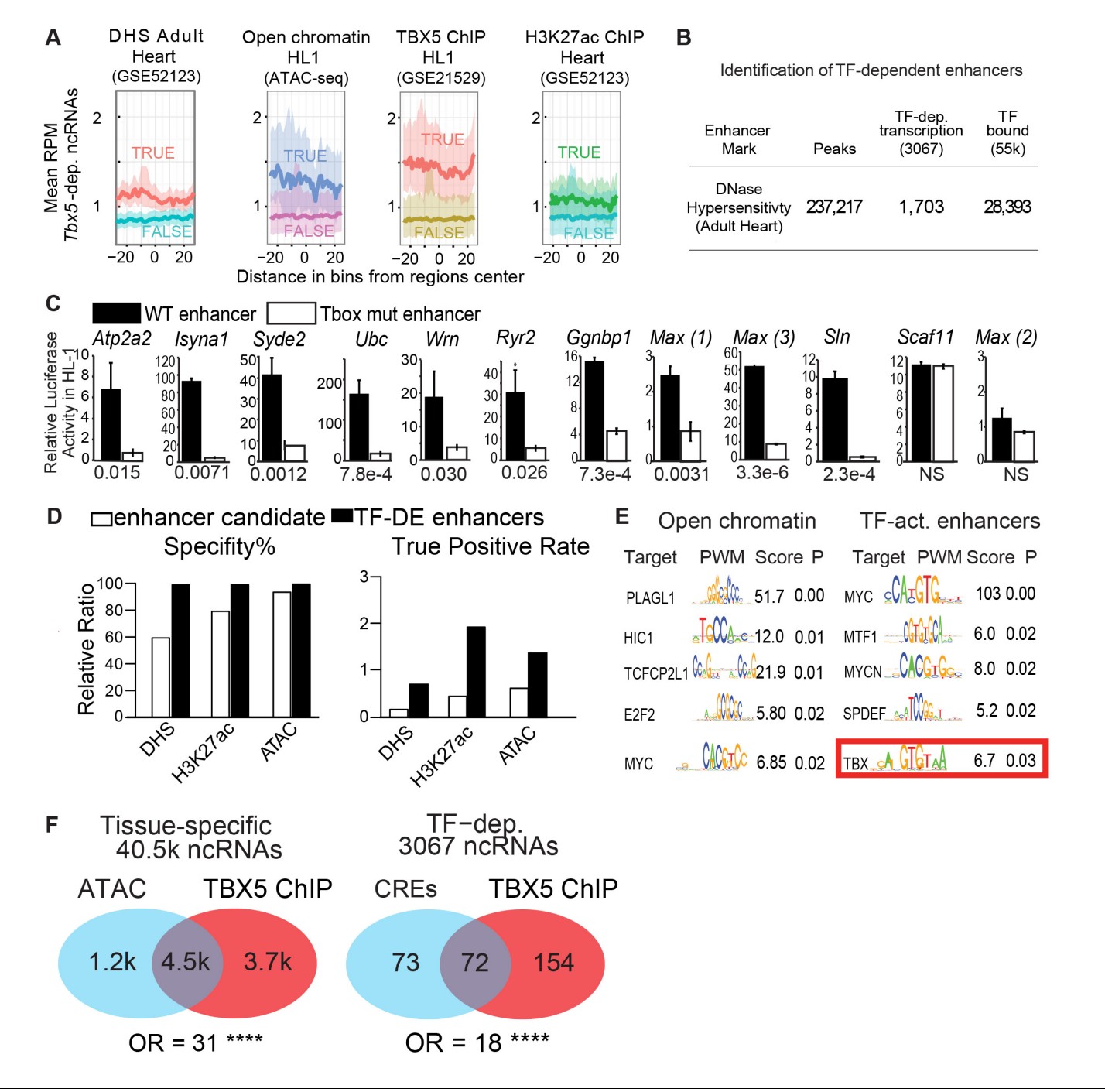

**Figure 2.** TF-dependent ncRNAs predict functional enhancers. (A) Meta-gene plot showing reads per million (RPM) of Tbx5-dependent ncRNAs, being divided by enhancer makers DNase Hypersensitivity (DHS), ATAC Sequencing in HL-1 cells, TBX5 ChIP-Seq, and H3K27Ac, respectively. Each subpanel is centered around all *Tbx5*-dependent ncRNAs, the averaged RPM of the ncRNAs hold an enhancer mark (top line, TRUE) vs those ncRNAs that do not (bottom line, FALSE). (B) Identification TF-dependent ncRNAs associated with cis-regulatory elements (CREs) by overlap with accessible chromatin defined by DNase hypersensitivity. Overlap of ncRNAs ("TF-dep transcription") and TF ChIP ("TF bound"), respectively. (C) Relative luciferase activity of select enhancers identified in HL-1 cardiomyocytes, labeled by nearest TBX5-dependent gene. Activity of wildtype enhancers (black) and T-box mutant enhancers (white) are shown. P-values for statistical comparisons of wildtype vs mutant enhancers are given below each graph. n = 3 each, activity normalized to co-transfected Renilla vector, and to vector with scrambled insert. (D) Predictive specificity and True Positive Rate of enhancer marks alone (white) and when combined with *Tbx5*-dependent ncRNA (black). Defined enhancers were identified and compared against 240 cardiac and 1162 non-cardiac enhancers reported in the VISTA database. The enhancers were defined by DNase hypersensitivity (DHS), H3K27 acetylation, and open

*Figure 2 continued on next page*

eLIFE Research article

Genes and Chromosomes | Genomics and Evolutionary Biology

*Figure 2 continued*

chromatin by ATAC-seq. (**E**) Motif analysis showing enriched motifs in HL-1 open chromatin (ATAC, left) and ncRNA-defined TBX5-dependent enhancers (right). Motif patterns are presented as position weight matrix (PWM) with enrichment statistics. Enrichment scores >1.5 and p-value <0.05 were considered significant, and the top five most-significant motifs are reported here. (**F**) Venn diagram of ATAC-seq and TBX5 ChIP-seq in background of ncRNAs (left) and TF-dependent elements overlapped with open chromatin in a background of TF-dep ncRNAs (right). Odds ratios (OR) are given for each background. **** indicates p<1e-10.

DOI: https://doi.org/10.7554/eLife.31683.007

strongly TBX5-dependent, as the activity of 10 of the 11 enhancers was significantly diminished by mutation of their canonical T-box sites (*Figure 2C*).

We asked whether differential ncRNA expression could discern functional TF-dependent enhancers from candidates marked by TF-ChIP or other marks of active regulatory elements. We examined the predictive power of TF-dependent ncRNA expression to identify accurately tissue-specific enhancers genome-wide, comparing the precision and specificity of *Tbx5*-dependent ncRNA-defined enhancers against those of 240 cardiac and 1162 non-cardiac enhancers from the VISTA database tested by mouse transient transgenic reporter assays *in vivo* (*Visel et al., 2007*). *Tbx5*-dependent transcription increased the predictive power of DNAse hypersensitivity (DHS), H3K27 acetylation, and open chromatin by ATAC-Seq (>2 fold for precision, with similar specificity, *Figure 2D*), suggesting that TF-dependent ncRNA expression can improve functional enhancer identification.

To further define atrial-specific TBX5-dependent enhancers, we overlapped *Tbx5*-dependent ncRNAs with open chromatin in the mouse atrium, defined by ATAC-Seq. From 16,000 open regions, we identified 145 local *Tbx5*-dependent ncRNAs marking 152 candidate *Tbx5*-dependent enhancers (empirical p<0.0001, *Figure 1—figure supplement 4*). We compared TF-motif enrichment between enhancers defined by *Tbx5*-dependent ncRNA versus all atrial open chromatin regions. Atrial open chromatin showed enrichment for motifs of TF families, such as MYC and E2F, that provide non-tissue-specific transcriptional functions (*Figure 2E*). By contrast, *Tbx5*-activated enhancers showed significant enrichment for the T-box motif itself (enrichment score = 6.7, p=0.03, PWMEnrich [*Frith et al., 2004*]). Furthermore, TBX5 occupancy by ChIP-seq in atrial HL-1 cells (*He et al., 2011*) was highly enriched at open atrial locations that contained *Tbx5*-dependent ncRNAs (72 of the 152 locations, OR = 18, p<2e-16; Fisher Exact Test, *Figure 2F*). Together, these observations provide evidence that TF-dependent enhancer transcription may be used to identify the locations of TF binding.

The candidate enhancers that were marked by differential transcription but not by TBX5 binding represent either indirect *Tbx5*-dependent elements or false negatives from ChIP. A candidate enhancer located at the *Sln* locus is one example: the region did not demonstrate TBX5 localization in ChIP-seq in HL-1 cells (*He et al., 2011*), but did show *Tbx5*-dependent ncRNA transcription. We interrogated this region by ChIP-qPCR and observed TBX5-occupancy (p=0.03 vs control *Gapdh* locus, two-tailed unequal variance t-test) (*Figure 3—figure supplement 1*). The element demonstrated robust TBX5-dependent activity in HL-1 cardiomyocytes in vitro as determined by luciferase assay (p = 2.3E-4 WT vs T-box Mutant [two-tailed unequal variance t-test], *Figure 2C*). Thus, we conclude that a regulatory element at the *Sln* locus, not previously identified by genomic approaches, is a direct TBX5-dependent enhancer, that has been identified by ncRNA Seq and validated by Chip-PCR.

## A quantitative relationship between *Tbx5*-dependent ncRNA and gene expression

We sought to define the gene regulatory networks that are controlled by *Tbx5*-dependent enhancer transcription and therefore examined enhancer–target gene interactions. We postulated a quantitative correlation between TF-dependent enhancer transcription and TF-dependent target gene expression. We observed that the direction of altered transcription of *Tbx5*-dependent ncRNAs and local polyA transcripts was most often coherent: 86% of downregulated (1577/1841; empirical p<0.001) and 66% of upregulated (1490/1769; empirical p=0.003) *Tbx5*-dependent ncRNAs were local to downregulated and upregulated *Tbx5*-dependent coding-genes, respectively (*Figure 1D,F*, *Figure 1—figure supplement 3B*). This significant directional coherence suggested a functional

relationship between *Tbx5*-dependent enhancer function and local *Tbx5*-dependent transcription. Targeted chromosome conformation capture analysis (4C-Seq) (*van de Werken et al., 2012*) was performed to identify enhancer–promoter contacts for four enhancers identified by differential ncRNA expression and validated in vitro. 4C-Seq in the atrial HL-1 cardiomyocyte cell line demonstrated significant interactions between the enhancers defined by *Tbx5*-dependent transcription and local *Tbx5*-dependent genes located within topological associated domains at the genetic locations of *Ryr2*, *Atp2a2*, *Ubc*, and *Wrn* (*Figure 3A*). 81 out of 93 enhancers with *Tbx5*-activated transcription showed quantitative directional coherence with the expression of the nearest *Tbx5*-dependent gene (by Pi-score [*Xiao et al., 2014*]) (*Figure 3B*), consistent with *cis*-regulation of the nearest TBX5-dependent target gene. We analyzed the fold-changes of *Tbx5*-dependent enhancer transcription with that of adjacent *Tbx5*-dependent genes and observed a significant correlation between their

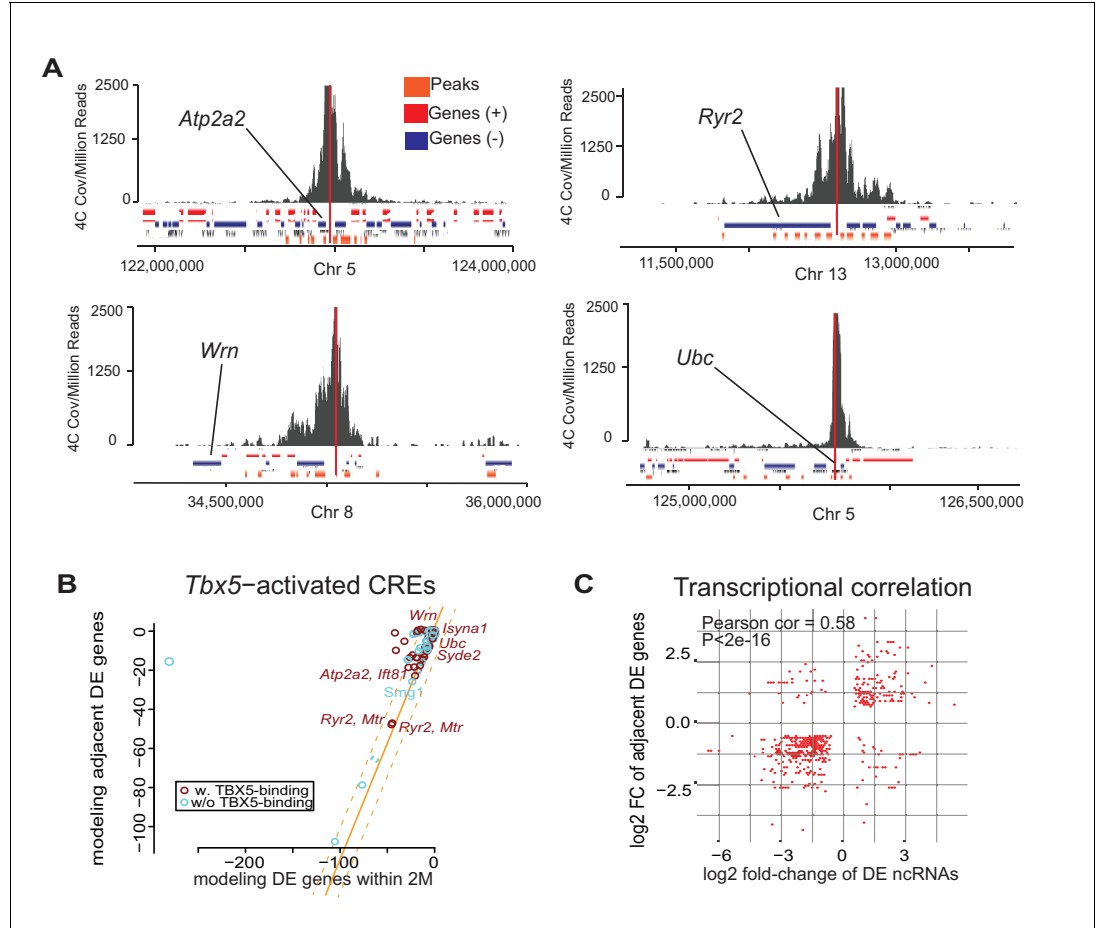

**Figure 3.** Enhancer transcription mediates target gene expression. (**A**) Genomic loci showing circularized chromatin conformation capture (4C) from the viewpoint of four identified regulatory elements (near *Atp2a2*, *Wrn*, *Ryr2*, and *Ubc*). Total reads (top) and significant contacts (bottom, orange) are plotted. Annotated genes are shown in blue and red (sense and antisense, respectively). The viewpoint is with a dashed red line and the nearest TBX5-dependent target is labeled. (**B**) Sparse scatterplot for combined Pi-scores calculated from *Tbx5*-activated enhancers and *Tbx5*-activated genes. The X-axis is the pi-score using the 'nearest' *Tbx5*-dependent gene, and the Y-axis is the pi-score using all potential TF-dependent gene targets within a 2-Mbp window. The orange line indicates y = x. The dashed lines mark the standard deviations. Example enhancers were labeled with the names of their nearest *Tbx5*-dependent gene. Red indicates enhancers with TBX5 occupancy, and blue indicates enhancers without TBX5 occupancy as determined by ChIP-seq. (**C**) Scatterplot in hexagonal binning for the 3067 identified TF-dependent ncRNAs. The X-axis is the differential expression fold change on a log$_2$ scale of these ncRNAs, and the Y-axis is the differential expression fold change on log$_2$ scale of any nearest gene targets. The fold change with the maximum absolute value is used when both neighbor genes are TF-dependent.

DOI: https://doi.org/10.7554/eLife.31683.008

The following figure supplement is available for figure 3:

**Figure supplement 1.** A TBX5-dependent enhancer at *Sln* that was not identified by TF ChIP-seq.

DOI: https://doi.org/10.7554/eLife.31683.009

direction and magnitude (coefficient = 0.58, p<2e-16 by Pearson Correlation test; *Figure 3B*). This positive correlation suggested that *Tbx5*-dependent enhancer transcription represents a quantitative metric describing *Tbx5*-dependent enhancer activity.

## A *Tbx5*-dependent ncRNA-defined enhancer at *Ryr2*

We hypothesized that *Tbx5*-dependent enhancer-associated lncRNAs may be functionally required for local *Tbx5*-dependent gene expression. To reveal functional gene sets within the identified cis-regulatory network, we performed Gene Ontology (GO) enrichment analysis (*Wang et al., 2015*) on the putative targets of enhancers that demonstrated *Tbx5*-activated transcription (*Figure 4A*). We found enrichment for two statistically significant terms, 'Calcium ion-related biological process' and 'Calcium ion transmembrane transport,' remarkably consistent with previous work demonstrating that *Tbx5* controls atrial rhythm by driving calcium-handling physiology and gene expression (*Nadadur et al., 2016*).

We next examined the enhancer at *Ryr2*, an essential calcium-handling gene component of the *Tbx5*-dependent calcium-handling GO term, for i*n-vivo* activity in zebrafish (*Figure 4B*). *Ryr2* directed cardiac expression of an EGFP reporter in zebrafish lines with a stably integrated transgene (*Figure 4B*, *Figure 4—figure supplement 2A–B*). Adult hearts displayed EGFP expression in cardiomyocytes lining the atrio- and bulbo-ventricular canals. To test whether expression activity was facilitated by T-box motifs, we mutated the seven canonical TBX5-binding sites located within the enhancer, which diminished the frequency of cardiac expression in $F_0$ zebrafish microinjected with the transgene construct (67/166 positive for wildtype vs 20/84 positive in the mutant and 5/95 for the *Fos* minimal promoter alone; p = 0.011, OR = 0.46 vs wildtype *RACER*, Fisher's Exact Test) (*Figure 4—figure supplement 2A–B*). Together, these results indicate that the enhancer at *Ryr2* is sufficient to drive expression in some cardiomyocytes *in vivo* and is promoted by TBX5.

We examined the relationship between *Tbx5*-dependent ncRNA abundance and enhancer activity at the *Ryr2* locus in more detail. We asked whether *Tbx5*-dependent ncRNA abundance could discern *Tbx5*-dependent enhancers among all candidate enhancers marked by TBX5-ChIP at *Ryr2*. Five candidate enhancers were defined by TBX5 ChIP at the *Ryr2* gene, which encodes the cardiac ryanodine receptor, a TBX5-dependent target that is critical for cardiac rhythm (*Nadadur et al., 2016*) (*Figure 4C*). The single candidate enhancer marked by significant *Tbx5*-dependent ncRNA abundance showed strong cardiac activity as determined by luciferase assay (*Figures 2C* and *4D*, p = 0.05, two-tailed unequal variance t-test). In contrast, the other candidates showed weaker or no activity in HL-1 cardiomyocytes (*Figure 4C,D*, p = 0.06, 0.39, 0.02 and 0.002 for Enhancers 1, 2, 3 and 5, respectively; two-tailed unequal variance t-test). This observation suggested that TF-dependent ncRNAs may improve the predictive performance of TF-occupancy-based regulatory element identification.

## *RACER*, a *Tbx5*-dependent lncRNA, is required for *Ryr2* expression

We examined the novel *Tbx5*-dependent lncRNA generated from the *Tbx5*-dependent enhancer at *Ryr2*. We refer to this lncRNA, located 25 kb upstream of the *Ryr2* locus, as Ryr2 Associated Cis-Element RNA (mm9 chr13:12226210–12226916, *RACER*, *Figure 4E*). We examined the functional requirement of *RACER* for *Ryr2* gene expression. We compared antisense-mediated knockdown of the enhancer-associated lncRNAs versus that of the mRNA in HL-1 cardiomyocytes (*Li et al., 2013*; *Ploner et al., 2009*). Control knockdown of the *Ryr2* mRNA caused a 50% reduction in *Ryr2* mRNA abundance. Enhancer ncRNA expression was unchanged (*Figure 4F*). By contrast, knockdown of the *Ryr2*-enhancer-associated ncRNAs (chr13:12226210–12226916 mm9) caused a 55% decrease in enhancer-associated ncRNA expression and, importantly, a corresponding 50% decrease in *Ryr2* mRNA abundance (*Figure 4F*). Knockdown of the control *Hprt* mRNA resulted in no significant change in the expression of *Ryr2* mRNA or RACER (*Figure 4—figure supplement 1*).

We examined the requirement for *RACER* for cardiomyocyte calcium kinetics. We measured the rate of calcium concentration increase using the calcium-sensitive dye Fluo4AM in spontaneously beating HL-1 cardiomyocytes. Calcium increase time represents a measure of RYR2 activity. In untransfected HL-1 cells, β-adrenergic stimulus (1 nM isoproterenol, ISO) caused a dramatic increase in the increase rate of cytosolic calcium transients (rate change from 33.3 ± 7.4 to 84.8 ± 9.9, p<0.05, two-tailed unequal variance t-test). HL-1 cardiomyocytes were transfected with antisense

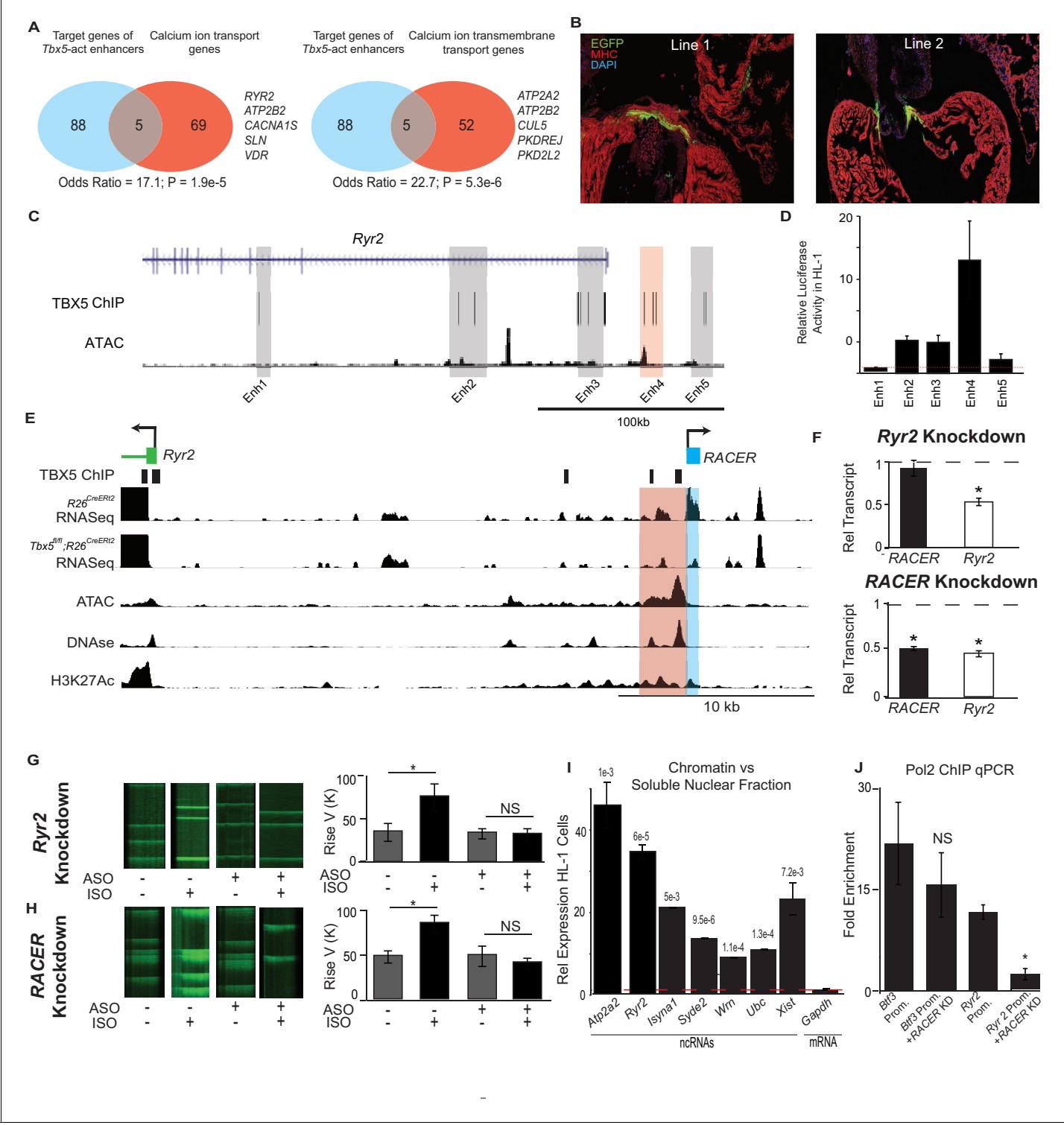

**Figure 4.** Enhancer-associated ncRNAs at *Ryr2* and *Atp2a2* are necessary for calcium-handling gene expression and cellular phenotype. (A) Gene Ontology functional enrichment for gene targets within a 2-Mb window of the TBX5-activated enhancers. Odds Ratio (OR) and P-value of overlap with GO terms 'Calcium Ion Transport Genes' and 'Calcium Ion Transmembrane Transport Genes'. (B) Representative 3 days post fertilization (Dpf) zebrafish embryos injected with a reporter construct containing the wildtype *Ryr2* enhancer reporter, showing cardiac EGFP expression. One Representative embryos from two stable lines are also shown (Line 1 and Line 2, one embryo each). Stable lines display EGFP fluorescence in the heart in addition to other tissues (see Supplement). (C) Genomic view of the *Ryr2* locus showing four candidate TBX5-dependent regulatory elements identified by TBX5 ChIP-seq (gray) and by *RACER* expression (Enh 4, red). (D) Relative luciferase activity in HL-1 cardiomyocytes of candidate *Ryr2* enhancers, normalized

*Figure 4 continued on next page*

*Figure 4 continued*

to co-transfected Renilla vector and to a vector with a scrambled insert (n > 3 replicates). (E) Genomic view of the *Ryr2* locus, showing identified noncoding RNA Ryr2 Associated Cis- Element RNA (RACER) in blue. A putative regulatory element associated with ncRNA is marked in red. Tracks show TBX5 ChIP (GSE21529), RNASeq from *R26^CreERt2* and *Tbx5^fl/fl*;*R26^CreERt2*, ATAC-Seq in HL-1 cardiomyocytes, DNASe hypersensitivity (ENCODE), and H3K27 acetylation (GSE52123). (F) Gene expression of *Ryr2* mRNA and the *RACER* transcript after knockdown of *Ryr2* mRNA (top) and knockdown of *RACER* (below). Relative transcript expression (RTE) after mRNA knockdown: 0.52 ± 0.03, p = 5.3E-5, for mRNA; RTE 0.92 ± 0.09, p = 0.43, and 0.82 for *RACER*. For ncRNA knockdown: RTE 0.44 ± 0.03 for mRNA and 0.45 ± 0.02 for *RACER*, p = 5.64E-6 and 0.04, respectively. (G, H) Representative calcium transient line scans from control (left) and ASO knockdown (right) HL-1 cardiomyocytes in the presence or absence of isopropteranol (+ISO, –ISO resp) after knockdown of *Ryr2* mRNA (top) or *RACER* (bottom) with antisense oligonucleotides (ASO). Quantification of rise velocity under control and isoproterenol treatments (right). (I) Relative transcript expression of chromatin enriched vs soluble nuclear fractions in HL1 cardiomyocyte cells of several ncRNAs and (as controls) Xist and the *Gapdh*-coding mRNA transcript. Expression was normalized to 18S ribosomal RNA. P values compared the relative expression in soluble vs nuclear fractions of the ncRNAs and Xist to that of GAPDH mRNA, and were less than Bonferroni corrected p<6e-3 for all ncRNAs. (J) Fold-enrichment of RNA Polymerase 2 (Pol2) occupancy by ChIP at housekeeping gene *Btf3* promoter (prom.) and at the *Ryr2* promoter in the control (left) and after antisense oligonucleotide knockdown of *Ryr2* enhancer ncRNA (ASO, right). Normalized to a control locus near the *Gapdh* gene.

DOI: https://doi.org/10.7554/eLife.31683.010

The following figure supplements are available for figure 4:

**Figure supplement 1.** Knockdown of control Hprt locus does not change *Ryr2* or *RACER* expression.

DOI: https://doi.org/10.7554/eLife.31683.011

**Figure supplement 2.** *Ryr2* enhancer shows cardiac expression in 3 days post fertilization (dpf) zebrafish.

DOI: https://doi.org/10.7554/eLife.31683.012

---

oligonucleotides (ASOs) specific to the *Ryr2* gene, or the *RACER* ncRNA, as described above. Control *Ryr2*-mRNA-knockdown blunted the effect of β-adrenergic stimulus (increase rate change from 34. 8 ± 5.9 to 33.3 ± 7.4 p = NS, Fisher's Exact Test) (*Figure 4G,H*). Knockdown of *RACER* blocked the effect of ISO, mimicking the effect of direct *Ryr2* knockdown. (increase rate change 49.3 ± 11.3 to 41.0 ± 6.1; p = NS, Fisher's Exact Test) (*Figure 4G,H*). We conclude that *RACER*, a *Tbx5*-dependent lncRNA at the *Ryr2* locus, is necessary for *Ryr2* gene expression and for normal *Ryr2*-dependent calcium kinetics, a cellular phenotype that is dependent on the *Tbx5*-driven gene regulatory network (*Nadadur et al., 2016*).

## *Tbx5*-dependent ncRNAs are chromatin-enriched

Enhancer-transcribed ncRNAs may function to stabilize enhancer–promoter contacts by association with chromatin (*Li et al., 2016*). We therefore determined the sub-nuclear localization of six *Tbx5*-dependent enhancer lncRNAs, including *RACER*, using analyses that compared the chromatin and soluble compartments from nuclear fractionation in HL-1 cardiomyocytes (*Werner and Ruthenburg, 2015*). We found that the *Tbx5*-dependent lncRNAs were all significantly enriched in the chromatin versus the soluble nuclear fraction (qPCR p<6e-3, n = 3 two-tailed unequal variance t-test) (*Figure 4I*). By contrast, control RNA from the *Gapdh*-coding region showed no chromatin enrichment. *Xist*, a canonical chromatin-binding ncRNA, served as a positive control and demonstrated a degree of chromatin enrichment similar to that of the examined *Tbx5*-dependent ncRNAs (*Figure 4I*). Thus, *Tbx5*-dependent enhancer-associated lncRNAs, and notably *RACER*, were tightly localized to chromatin, suggesting a possible direct role in the regulation of their target genes (*Li et al., 2013*; *Lai et al., 2013*).

We examined the necessity of *RACER* for RNA Polymerase 2 (Pol2) occupancy at the *Ryr2* promoter as a possible mechanism for its requirement for *Ryr2* gene expression. Knockdown of *RACER* caused a quantitatively significant fall in Pol2 occupancy at the *Ryr2* promoter, suggesting that this enhancer-bourne lncRNA is necessary for recruiting or stabilizing Pol2 at the target gene (Pol2 ChIP enriched vs ctrl locus 8.7 ± 0.79 fold vs 1.6 ± 0.20 fold in knockdown, p = 0.02, two-tailed unequal variance t-test, *Figure 4J*). There was no change in Pol2 occupancy at the housekeeping gene *Btf3* after *RACER* knockdown (21.3 ± 5.99 fold vs 14.3 ± 4.85 fold, p = 0.46, two-tailed unequal variance t-test, *Figure 4J*).

## Discussion

### TF-dependent enhancer ncRNAs define a TF-dependent gene regulatory network

Defining TF-dependent gene regulatory networks (GRNs) that include TF-dependent enhancers in health and disease remains a fundamental challenge. Here, we analyze *Tbx5*-dependent ncRNAs to define *Tbx5*-dependent enhancers in a sensitive, quantitative manner. Taken in context, this work contributes to the concepts that enhancer-bourne ncRNAs (1) mark active *cis*-regulatory elements and (2) are essential for *cis*-regulatory element function (*Figure 5*). Even though we are unsure of the precise mechanism, this phenomenon may be used as a reliable metric of active enhancers. The differential ncRNA approach appears to provide non-redundant data when compared with other current state-of-the-art methods that attempt to define enhancers by proxy, using genomic accessibility (e.g. DNAse-seq or ATAC-seq) or histone status (e.g. H3K27ac). The quantitative correlation between the RNA abundance of the enhancer and that of the target promoter that we observed (*Figure 3C*) suggests that differential TF-dependent ncRNA-seq may provide a direct readout of differential enhancer function (*Figure 5*). The quantitative relationship between enhancer ncRNA and target promoter mRNA abundance may have been underestimated, based on the assumption that *Tbx5*-dependent enhancer ncRNAs function at the nearest *Tbx5*-dependent promoters. Future integration of chromosome architecture for the resolution of enhancer-promoter contacts may improve this relationship. Furthermore, efforts to define the mechanisms of action of enhancer-generated ncRNAs will more precisely define the quantitative relationship between the transcriptional output of enhancers and that of their target promoter.

### The mechanism of action of TF-dependent enhancer ncRNAs

TF-dependent lncRNAs such as RACER may be functionally required components of the TF-driven gene regulatory network. The *Tbx5*-dependent enhancer-associated lncRNA *RACER* is required for *Ryr2* expression, tightly associated with chromatin, and required for the recruitment or stabilization of Pol2 binding at the *Ryr2* promoter (*Figure 4I*). Several lncRNAshave been shown to promote the transcriptional activation of coding genes at a distance, and the mechanisms by which lncRNAs achieve their implicated roles has become a central question (*Wang et al., 2011*; *Rinn et al., 2007*; *Davidovich and Cech, 2015*; *Engreitz et al., 2016*). Chromatin associated RNAs (cheRNAs) are a recently described class of lncRNAs that are biochemically defined by tight chromatin-engagement as a consequence of their ongoing transcription (*Werner and Ruthenburg, 2015*). We observed that all validated *Tbx5*-dependent ncRNAs were also cheRNAs (*Figure 4I*). CheRNAs have been shown to act locally to promote proximal gene transcription in a tissue-specific fashion (*Werner and Ruthenburg, 2015*; *Werner et al., 2017*), consistent with the correlation between *Tbx5*-dependent CRE-associated ncRNA expression and *Tbx5*-dependent nearest gene expression. The dependence

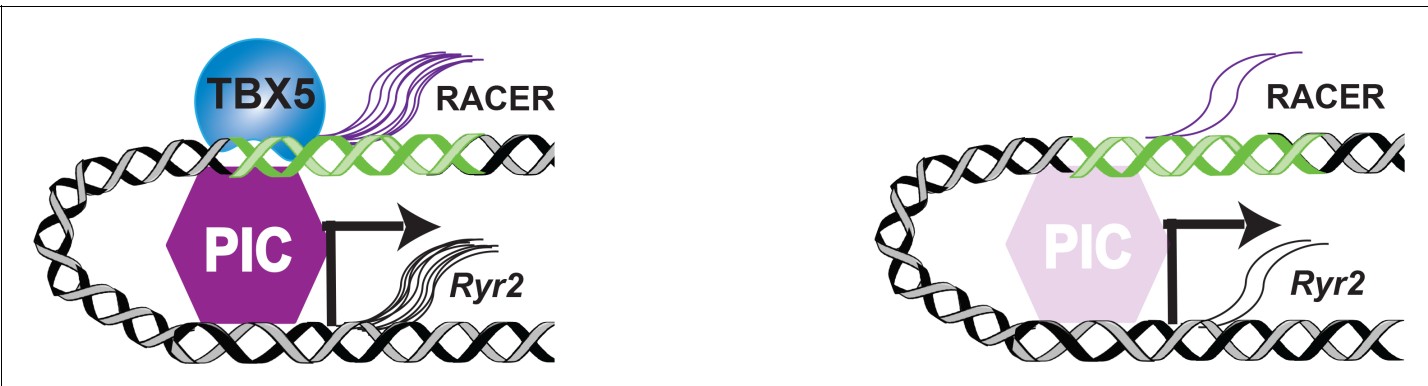

**Figure 5.** TF-dependent transcription of regulatory elements. mRNA and enhancer transcription occur in the presence of transcription factor (TBX5) through the Transcription Preinitiation Complex (PIC) (left). Loss of the TBX5 results in decreased PIC complex, leading to decreased enhancer ncRNA (RACER) and mRNA (*Ryr2*) transcription (right).
DOI: https://doi.org/10.7554/eLife.31683.013

of target gene transcription on enhancer transcription may be a fundamental aspect of TF-driven cis-regulatory networks (*Figure 5*).

### *Tbx5*-dependent ncRNA-defined enhancers: a new class of enhancers for cardiac rhythm

Genome-wide approaches to defining gene regulatory networks are essential for the interpretation of human genetic analysis using whole genome sequencing or genome-wide association studies (GWAS). In both cases, ascertaining the functional impact of noncoding single nucleotide polymorphisms (SNPs) is challenging. Novel approaches for the discovery of functionally relevant enhancers on a genome-wide scale are necessary, and recently described approaches have made strides in this arena (*Arnold et al., 2013*; *Kwasnieski et al., 2012*; *Patwardhan et al., 2012*). We applied TF-dependent ncRNA-sequencing to define TBX5-driven enhancers, based on a priori knowledge that TBX5 drives a gene regulatory network for cardiac rhythm homeostasis (*Nadadur et al., 2016*). We expected that the TBX5-driven enhancers that we identified would help to define a cardiac rhythm network, and consistently, *Tbx5*-dependent ncRNA-defined enhancer targets were highly enriched in GO terms critical for cardiac rhythm (*Figure 4*). Future efforts will determine whether arrhythmia-associated SNPs are enriched in human enhancers identified by TF-dependent ncRNAs. This approach may contribute to efforts to define the genomic locations and mechanisms underlying noncoding human genetic signals.

## Materials and methods

### Identifying open chromatin

ATAC-seq library preparation and sequencing

We cultured HL-1 cardiomyocytes according to published protocols (*Neilan et al., 2000*). ATAC (assay for transposase-accessible chromatin) protocol was performed as previously described (*Buenrostro et al., 2015*). Briefly, nuclei were isolated according to published protocols, transposed, and libraries were amplified and normalized with the Illumina Nextera DNA Library prep kit (FC-121–1031) according to the manufacturer's protocols and the methods published by *Buenrostro et al. (2015)*. Libraries were quantitated by the Agilent Bio-analyzer, pooled in equimolar amounts, and sequenced with 50-bp single-end reads on the HiSeq2500 following the manufacturer's protocols.

Data analysis

The 50-bp reads were examined by FastQC. About 30–70 million high-quality reads (quality score >30) were sequenced for each sample. All reads for each sample were aligned to the UCSC mouse genome mm9 with Bowtie (v1.0.0) using default settings except (-v 1 -q -m 1). Duplicates, low-quality (Q < 30), and chrM reads were removed, and an index was added by samtools (v 0.1.18). Between 12 million and 26 million high-quality reads per sample that mapped to genomic DNA were retained. Peaks were called for each sample using MACS2 (*Zhang et al., 2008*) with the parameters '–nomodel –bw 200 –bdg'. Overlap peaks identified from three replicates were generated for the downstream analysis. Peaks that mapped to the ENCODE blacklist regions were excluded from the identification. Peak annotation was performed using the Bioconductor package seq2pathway (*Wang et al., 2015*). All sequencing tracks were viewed using the Bioconductor package Gviz (*Hahne and Ivanek, 2016*).

### Differentially expressed coding genes identified from RNA-seq analysis

Mouse model

Mice were raised according to university protocols (protocol # 71737). A tamoxifen-inducible Cre recombinase driven by the ubiquitous R26 promoter was used to excise exon 3 of *Tbx5* as previously described (*Arnolds et al., 2012*). Excision of *Tbx5* was induced by IP tamoxifen for 3 days, 0.167 mg tamoxifen/day. Mice were sacrificed at 1 week post tamoxifen and right and left atrial appendages were dissected.

## RNA extraction

The left atria of these 6-8 weeks old mice were mechanically homogenized in TRIzol Reagent (Invitrogen), chloroform was added, the aqueous phase was isolated, ethanol was added, and then RNA was isolated using RNeasy Mini Column (Qiagen).

## RNA-Seq library preparation

Libraries were prepared from this RNA starting with 1 g per sample and using the mRNA-seq Sample Prep Kit (Illumina) as per recommended instructions. After Ribozero purification and removing only ribosomal RNA, barcoded libraries were prepared according to Illumina's instructions (2013) accompanying the TruSeq RNA Sample prep kit v2 (Part# RS-122-2001). Libraries were quantitated using the Agilent Bio-analyzer (model 2100) and pooled in equimolar amounts. The pooled libraries were sequenced with stranded 50-bp single-end reads on the HiSeq2500 in Rapid Run Mode following the manufacturer's protocols (2013).

## Data analysis

The left 3 bp of the sequencing reads were trimmed due to their unexpected lowest quality score among all the 50 bp reads. About 62–72 million high-quality reads (quality score >30) for each sample were obtained. Fastq files were aligned to build GRCm38.p3 of the mouse genome provided by the GENCODE Consortium (mm10). Transcript alignment was performed using TopHat (version 2.0.10, using the parameters –segment-length 24 –segment-mismatches 2 –no-coverage-search). Reads mapping to the mitochondrial genome were excluded. Between 43 million and 48 million successfully mapped reads were analyzed for differential expression using the R package DESeq (*Anders et al., 2013*). Significance was considered when FDR was <0.05 and fold change was >1.5. To ensure cardiac expression, we additionally required a normalized mean value in the widetype group >1.

## Differentially expressed noncoding RNAs identified from RNA-Seq analysis

### RNA-Seq library preparation

Total RNA was extracted by TRIzol Reagent (Invitrogen), followed by ribosomal and polyA depletion. After RiboZero purification and oligo-dT depletion, RNA Barcoded Libraries were prepared according to Illumina's instructions (2013) accompanying the TruSeQ RNA Sample prep kit v2 (Part# RS-122-2001). Libraries were quantitated using the Agilent Bio-analyzer (model 2100) and pooled in equimolar amounts. The pooled libraries were sequenced with 50-bp stranded single-end reads on the HiSEQ4000 in Rapid Run Mode following the manufacturer's protocols (2013).

### Data analysis

The left 3 bp of the fastq sequencing reads were trimmed due to their unexpected lowest quality score among all of the 50-bp reads. About 92–135 million high-quality reads (quality score >30)reads for each sample were obtained. Fastq files were aligned to build GRCm38.p3 of the mouse genome provided by the GENCODE Consortium (mm10). Transcript alignment was performed using TopHat (version 2.0.10) as previously described (*Trapnell et al., 2009*) and between 89 million and 129 million reads were successfully mapped. De novo assembly was performed by Cufflinks (version 2.2.1, with parameters –g –frag-bias-correct –multi-read-correct –upper-quartile-norm), as it can recover transcripts that are transcribed from segments of the genome that are missing from the current genome assembly. Featurecount was performed using Rsubread. Analysis of differential expression was performed using a count-based differential expression analysis strategy for RNA sequencing and normalized using the R package DESeq (*Anders et al., 2013*). False discovery rate (FDR) was calculated after removing the coding-gene counts. Significance was considered to have been reached when FDR was <0.05 and fold change was >1.5. We defined unidirectional transcripts as those transcribed from one strand and bi-directional transcripts as those transcribed from both strands. The mm10 genomic coordinates of identified noncoding transcripts were lifted over to the mouse mm9 genome when overlay with identifeid or published enhancer regions.

## Associating ncRNAs and regulatory elements

We observed that around half of the identified TF-dependent ncRNAs were contiguous to rather than directly overlapping with called open chromatin regions. To capture these ncRNAs, we expanded the definition of 'co-localization' of ncRNAs and enhancers to include sequences that reside with less than 1 kb between their centers of regions. Compared to edge-to-edge distance control, this strategy counts two adjacent short regions but excludes two adjacent long regions. As longer regions have more chance to overlap with each other directly, this strategy adjusted the bias introduced by the width of the called peaks.

## Simulation study on ncRNAs local to coding genes that were TF-dependent

The null hypothesis is that TF-dependent ncRNAs and genes are randomly localized across the whole genome. To test this hypothesis, we ran $n = 1000$ simulations. In each simulation run $i$, we randomly sampled $x$ ncRNAs from the 40.5K noncoding transcript background and $y$ genes from the mouse genome, termed set $\{X\}$ and set $\{Y\}$, respectively. We then counted the number of ncRNAs in the set $\{X\}$ within the 2-Mb window of any genes within set $\{Y\}$, and recorded it as $a_i$. A is the observed count of TF-dependent ncRNAs within the 2-Mb window of TF-dependent genes. An empirical p-value was estimated for TF-activated and TF-repressed ncRNAs, respectively (*Equation 1*), let the expression marked by |.| be the total number of counts meeting a given criteria.

$$Pvalue = \frac{\sum_{i=0}^{n} |a_i \geq A|}{n} \tag{1}$$

## Simulation study on open chromatin regions that were TF-dependent

To test the 3067 TBX5-dependent ncRNAs for enrichment in chromatin open regions, we randomly sampled 3067 regions from the background of total identified chromatin open loci from atrial HL-1 cells (16,000). We estimated how many regions would be Tbx5-dependent enhancers ($k$) by chance alone. We repeated the simulation n (10,000) times, and each time, $k$ estimated from the simulation (range from 13 to 68) was far less than the observed value A (161, *Figure 1—figure supplement 4*). This provided an empirical p-value of p<0.0001(*Equation 2*).

$$Pvalue = \frac{\sum_{i=0}^{n} |k_i \geq A|}{n} \tag{2}$$

Specifically in this simulation, we allowed a separation of 500 bp between the ATAC peak and noncoding transcripts to define overlap. We observed 161 candidate *Tbx5*-dependent enhancers (using the R function subsetByOverlaps). Note that in rest of this study, we overlapped chromatin accessible regions (DNAse or ATAC) with noncoding transcripts as those within a 1-kb windowfrom peak-centers (using an in-house Python script), which reported 152 *Tbx5*-dependent enhancers. The subsetByOverlapsends function with a separation of 500 bp found all of the regions that the peak-centers strategy found, but the subsetByOverlapsends function is more friendly to perform and thus was applied for these simulations.

## Combined Pi-score to quantify significance of differential expression

To prioritize differentially expressed genes together with their cis-regulatory elements, we designed a computational method called a 'combined Pi-score'. Previously, the Pi-score has been proposed to improve the prioritization of differentially expressed genes by combining expression fold-change and statistical significance (i.e. the p-value) (*Xiao et al., 2014*). We modified the original Pi-score to fit the current data distribution (*Equation 2*).

$$\pi_i = \varphi_i \cdot (-log_{10}P_i) \tag{3}$$

where side effect $\varphi_i$ was either the fold-change when fold-change is finite, or was an assigned value $\pm X$ representing the edging values of all finite fold-changes when fold-change is infinite. Here, we averaged the Pi-score of a ncRNA $i$ and the Pi-score of its gene target to calculate the combined Pi-score (*Equation 3*). In this study, different strategies to select the target genes of an ncRNA were compared.

$$c\boldsymbol{\pi}^i = \left[\boldsymbol{\pi}^i_{ncRNA} + \max\left(\boldsymbol{\pi}^i_{target}\right)\right]/2 \tag{4}$$

Briefly, we calculated the Pi-score for both the noncoding transcriptome and the coding transcriptome and compared the combined Pi-scores of two models of ncRNA targeting. The 'nearest mapping' model assumes that ncRNAs identify enhancers that regulate the nearest target gene, the 'broader mapping' model assumed that all potential TBX5-dependent target genes within a window of 2 Mb could be the targets of an enhancer.

## Motif enrichment analysis

We tested the motif enrichment for the identified enhancers as well the whole set of chromatin-accessible regions in HL1 cells. 637 pre-compiled threshold-free position weight matrices (PWM) for mouse from the Bioconductor package MotifDb were downloaded for testing. Motif enrichment was calculated using the threshold-free affinity scoring algorithm with a lognormal distribution background (*Frith et al., 2004*). Significance was considered when p<0.05 and the enrichment score was >1.

## Functional enrichment of gene targets of enhancers

A background corrected Fisher's Exact test was performed using the Bioconductor package Seq2-pathway (*Wang et al., 2015*). The functional enrichment test was examined against 253 Gene Ontology Biological Process (BP) terms that have directevidence and are annotated to less than 1000 gene members. Significance was considered when FDR was <0.05 and the score of the odds ratio was >1.

## Landscape for mouse heart enhancer marks

The genome-wide landscape of H3K27Ac in whole mouse adult heart was called by macs2 and shared by the authors (GSE52123) (*He et al., 2014*). Two biological replicates and 44,707 regions were commonly marked by H3K27Ac. The footprinting for DNase I hypersensitive sites in whole mouse adult heart was downloaded from ENCODE/University of Washington (*Stergachis et al., 2014*). Transcription factor and P300 landscapes in HL1 cardiomyocytes were downloaded from GEO (GSE21529) (*He et al., 2011*).

## Retrieving phastCons sequence conservation scores

The phastCons30 sequence conservation scores were retrieved by querying against the UCSC Table Browser using the Bioconductor package rtracklayer (*Lawrence et al., 2009*). An average score was calculated for each selected region, removing the NA values. For certain large classes of genomic regions, such as lincRNAs or open chromatin regions, we randomly sampled 2000 regions first, and then retrieved the conservation scores for each sampled region to represent the features of the whole class.

## Meta-gene analysis

To compare the accessibility of chromatin at selected regions for groups of features (*Figure 1A*), meta-gene analysis was performed using the Bioconductor package metagene (v 2.2.1). Multiple combinations of groups of features were compared in a single analysis, and BAM files instead of the peaks produced by peak callers were used to increase the sensitivity. Briefly, four steps were performed for meta-gene analysis. (1) The read coverages of all selected regions were extracted from BAM files and normalized to reads per million aligned (RPM). (2) We binned the position of each selected region with a size parameter of 50 bp, and calculated the averaged RPM within each bin. (3) We estimated the confidence interval of each binned position using bootstrapping. (4) And finally, we plotted the mean value as a line and the estimation as a ribbon.

The same analysis was also performed to compare the TF-dependent expression at selected regions for groups of features (*Figure 3A*).

## 4C experiments

4C-seq was performed as described previously (*van de Werken et al., 2012*). In short, HL-1 cardio-myocytes were collected and cross-linked with 1% formaldehyde in PBS with 10% FCS for 10 min at room temperature, nuclei were isolated and cross-linked DNA was digested with a primary restriction enzyme recognizing a 4-bp restriction site (DpnII), followed by proximity ligation. Cross-links were removed and a secondary restriction enzyme digestion (Csp6I) initiated, followed again by proximity ligation. For all experiments, 200 ng of the resulting 4C template was used for the subsequent PCR reaction, of which 8 samples (total: 1.6 μg of 4C template) were pooled and purified for next-generation sequencing.

Sequencing reads were aligned using bowtie2 (v2.2.5 with parameter -q set to 1) and processed as described previously (*van de Werken et al., 2012*). In short, reads are mapped to a restricted mouse reference genome (mm9), consisting of sequences directly flanking the 4C primary restriction enzyme site (DpnII), called 4C frag-ends. Non-unique frag-ends are discarded in subsequent analysis. After mapping, the highest-covered frag-end was removed from the dataset in the normalization process and data were read-depth normalized to 1 million aligned intrachromosomal reads. 4C-Seq coverage profiles were calculated as 'running means', i.e. coverage averages of 21 consecutive 4C frag-ends.

## Nuclear fractionation for assessing chromatin enrichment

Fraction of HL-1 nuclei was performed as previously described *Werner and Ruthenburg, 2015*. In short, ~15 million cells from three different populations were lysed with 0.1% Triton X-100 and nuclei were pelleted 1300 x g for 12 min, 4°. Nuclei were re-suspended in 250 μl NRB (20 mM HEPES pH 7.5, 50% glycerol, 75 mM NaCl, 1 mM DTT, 1 x protease inhibitor cocktail), and then an equal volume of NUN buffer (20 mM HEPES, 300 mM NaCl, 1 M Urea, 1% NP-40 substitute, 10 mM $MgCl_2$, 1 mM DTT) was added and cells were incubated for 5 min on ice. After centrifugation (1200 x g, 5 min, 4°C), RNA from the supernatant (soluble nuclear extract) and chromatin (pellet) were extracted using Trizole, purified on Zymo RNA Clean and Concentrator columns, and then converted to cDNA using random hexamer primers and High Performance MMLV Reverse Transcriptase from Epicentre. qPCR was then performed using SYBR green master mix (Life Technologies), with 18S rRNA (F primer: CGCAGCTAGGAATAATGGAATAGG, R primer: GCCTCAGTTCCGAAAACCAA) as an internal standard for both fractions.

Quantitative PCR was performed using the Power SYBR Green master mix (Life Technologies) on a CFX384 Touch Real-Time PCR Detection System (BIO-RAD). qPCR data were analyzed using the ΔΔCt method. Averages (arithmetic mean) of target noncoding RNA Ct values from three or more technical replicates were normalized to the geometric mean of the reference genes *eEF2*, *LRC8*, and *eIF3 subunit 5* (*Kouadjo et al., 2007*). For each independent sample, the resulting ΔCt values were used to compute fold-changes between the chromatin-enriched and soluble nuclear fractions (ΔΔCt). Error bars correspond to uncertainty in the chromatin pellet extract ΔCt values used for fold chromatin enrichment calculation, reflecting the only variance from qPCR technical replicates, calculated as the quadrature sum of the target noncoding RNA Ct values' standard deviation and uncertainty in the normalization reference value. This latter uncertainty was calculated as the quadrature sum of the standard error of each reference gene's raw Ct value distribution across qPCR technical replicates.

## Relative luciferase activity

Luciferase response assays were performed as previously described (*Nadadur et al., 2016*). Candidate regulatory elements were amplified from C57/B6 mouse genomic DNA or synthesized via Gibson Assembly. Sequence was verified and then cloned into the pGL4.23 enhancer luciferase response vector with a minimal promoter. HL-1 cardiomyocytes were co-transfected with luciferase response vector and a pRL control using Lipofectamine 3000, cultured for 48 hr after transfection, then lysed and assayed using the Dual-Luciferase Reporter Assay system (Promega).

## Transgenic zebrafish

Mouse *RACER*-associated sequences were cloned upstream of the mouse minimal *c-fos* promoter and *EGFP*. Fertilized one-cell zebrafish embryos were microinjected with DNA and Isce I as

described previously (Goldman et al., 2017). To test whether expression patterns were due to insertional effects, we analyzed two independent stable lines, finding that they had similar expression patterns. The full name of the transgenic lines are Tg(mRyrENfos:EGFP-L1)$^{pd305}$ and Tg(mRyrENfos:EGFP-L2)$^{pd306}$. For predicted TBX5-binding sites, we compared wildtype enhancers or enhancers with the seven T-box motifs mutated with a construct containing the minimal promoter driving EGFP alone. Fluorescence was assessed in ($F_0$) after microinjection at 3 days post fertilization (dpf).

### Antisense-mediated knockdown

Custom antisense oligonucleotides (ASOs) were designed and synthesized by IDT. Pooled ASOs were transfected with Lipofectamine 3000 into HL-1 cardiomyocytes at a concentration of 10 nM. Cells were sorted into transfected and non-transfected populations by flow cytometry based on co-transfection with a vector encoding mCherry. ncRNA expression, Ryr2 coding transcript, and calcium kinetics were measured. ASOs against the Ryr2 lncRNA were: 5'-CCCTTCATATCACGTTGGAA-3', 5'-ATCACCTGCCCTGGTTCTTT-3', 5'-TATGATCACCTGCCCTGGTT-3' and 5'-CTTCATGCTTATCTGACAAG-3'. Antisense oligos against the Ryr2 mRNA were 5'-CGATGTCCTTGGCAGGCTCA-3', 5'-CATCGTCCATGTGGCCTTCG-3', 5'-ATTCATCCATGTGCCCATGC-3', 5'- GTCTGTCCCACTGGCCTTTG −3', and 5'- ATGTCTGGCGTGTCACCATC −3'.

### Calcium transient measurement

Cytosolic [$Ca^{2+}$]$_i$ was measured with the high-affinity $Ca^{2+}$ indicator Fluo-4 (Molecular Probes/Invitrogen) using a laser-scanning confocal microscope (Zeiss LSM 510) equipped with a $\times$ 63/1.40 NA oil-immersion objective lens. HL-1 cells were sorted and plated on coated coverslips and incubated at room temperature with 10 μM Fluo-4/AM for 15 min in normal Tyrode's solution containing: 140 mM NaCl, 4 mM KCl, 10 mM glucose, 10 mM HEPES, and 1 mM $MgCl_2$, 1 mM $CaCl_2$, with pH 7.4 using NaOH, followed by a 10 min perfusion wash with Tyrode (at 37 ˚C) without Fluo-4 before recording. Fluo-4 was excited with the 488 nm line of an argon laser and fluorescence was measured at >515 nm. Spontaneous $Ca^{2+}$ transients were acquired in line-scan mode (3 ms per scan; pixel size 0.12 μm). $Ca^{2+}$ transients are presented as total fluorescence intensity normalized to resting fluorescence ($F/F_0$) during steady-state conditions before field stimulation. Cells were recorded at baseline, then isopropteranol was perfused to a final concentration of 1 nM and calcium transients were recorded again.

## Acknowledgements

This research was supported in part by NIH R01 HL092153, R01HL124836, and R33 HL123857 to IPM, R21LM012619 to XHY and F30 HL131298 to RDN, by an AHA Collaborative Sciences Award to IPM and AR (NIH R01 HL 081674) and by an AHA Merit Award to KDP. This research was supported in part by the Leducq Foundation (IPM), and by the NIH through resources provided by the Computation Institute and the Biological Sciences Division of the University of Chicago and Argonne National Laboratory, under grant 1S10OD018495-01. We specifically acknowledge the assistance of Lorenzo Pesce.

## Additional information

### Funding

| Funder | Grant reference number | Author |
| --- | --- | --- |
| National Institutes of Health | | Xinan H Yang<br>Rangarajan D Nadadur<br>Ivan P Moskowitz |
| National Institutes of Health | R21LM012619 | Xinan H Yang |
| National Institutes of Health | F30HL131298 | Rangarajan D Nadadur |
| Fondation Leducq | | Wouter de Laat<br>Ivan P Moskowitz |

| | | |
|---|---|---|
| American Heart Association | | Ivan P Moskowitz<br>Alexander J Ruthenburg |
| National Institutes of Health | R01 HL092153 | Ivan P Moskowitz |
| National Institutes of Health | R01HL124836 | Ivan P Moskowitz |
| National Institutes of Health | R33 HL123857 | Ivan P Moskowitz |

The funders had no role in study design, data collection and interpretation, or the decision to submit the work for publication.

## Author contributions

Xinan H Yang, Resources, Data curation, Software, Formal analysis, Supervision, Funding acquisition, Validation, Investigation, Visualization, Methodology, Writing—original draft, Project administration, Writing—review and editing; Rangarajan D Nadadur, Conceptualization, Resources, Data curation, Software, Formal analysis, Supervision, Validation, Investigation, Visualization, Methodology, Writing—original draft, Project administration, Writing—review and editing; Catharina RE Hilvering, Joseph Aaron Goldman, Data curation, Formal analysis, Validation, Investigation, Visualization, Methodology, Writing—original draft, Writing—review and editing; Valerio Bianchi, Conceptualization, Resources, Data curation, Software, Formal analysis, Validation, Investigation, Visualization, Methodology, Writing—original draft, Writing—review and editing; Michael Werner, Conceptualization, Data curation, Formal analysis, Investigation, Methodology, Writing—original draft, Writing—review and editing; Stefan R Mazurek, Formal analysis, Validation, Investigation, Visualization, Methodology, Writing—original draft, Writing—review and editing; Margaret Gadek, Formal analysis, Validation, Investigation, Visualization, Writing—review and editing; Kaitlyn M Shen, Data curation, Formal analysis, Validation, Investigation, Writing—review and editing; Leonid Tyan, Jenna Bekeny, Data curation, Formal analysis, Validation, Investigation, Visualization, Writing—review and editing; Johnathon M Hall, Data curation, Formal analysis, Validation, Investigation, Visualization, Methodology, Writing—review and editing; Nutishia Lee, Data curation, Formal analysis, Validation, Investigation, Visualization, Methodology; Carlos Perez-Cervantes, Resources, Software, Validation, Investigation, Visualization, Methodology, Writing—review and editing; Ozanna Burnicka-Turek, Data curation, Investigation, Methodology, Writing—review and editing; Kenneth D Poss, Resources, Data curation, Formal analysis, Supervision, Funding acquisition, Validation, Visualization, Methodology, Writing—review and editing; Christopher R Weber, Software, Formal analysis, Validation, Investigation, Visualization, Methodology, Writing—original draft, Writing—review and editing; Wouter de Laat, Conceptualization, Data curation, Formal analysis, Funding acquisition, Validation, Investigation, Visualization, Methodology, Writing—original draft, Project administration, Writing—review and editing; Alexander J Ruthenburg, Conceptualization, Resources, Data curation, Software, Formal analysis, Funding acquisition, Validation, Investigation, Visualization, Methodology, Writing—original draft, Project administration, Writing—review and editing; Ivan P Moskowitz, Conceptualization, Resources, Data curation, Software, Formal analysis, Supervision, Funding acquisition, Validation, Investigation, Visualization, Methodology, Writing—original draft, Project administration, Writing—review and editing

## Author ORCIDs

Valerio Bianchi http://orcid.org/0000-0001-9025-7923
Alexander J Ruthenburg http://orcid.org/0000-0003-2709-4564
Ivan P Moskowitz https://orcid.org/0000-0003-0014-4963

## Ethics

Animal experimentation: Studies were performed according to university protocols, approved by the Institutional Animal Care and Use Committee (protocol # 71737)

## Decision letter and Author response

Decision letter https://doi.org/10.7554/eLife.31683.020
Author response https://doi.org/10.7554/eLife.31683.021

# Additional files

## Supplementary files

• Transparent reporting form
DOI: https://doi.org/10.7554/eLife.31683.014

## Major datasets

The following previously published datasets were used:

| Author(s) | Year | Dataset title | Dataset URL | Database, license, and accessibility information |
|---|---|---|---|---|
| He A, Gu F, Hu Y, Ma Q, Ye L, Visel A, Pennacchio LA, Pu WT | 2014 | Reinstatement of developmental stage-specific GATA4 enhancers controls the gene expression program in heart disease | https://www.ncbi.nlm.nih.gov/geo/query/acc.cgi?acc=GSE52123 | Publicly available at the NCBI Gene Expression Omnibus (accession no: GSE52123) |
| Kong S, He A, Pu WT | 2011 | Cardiac transcription factors in HL-1 cells: gene expression and genome binding profiling | https://www.ncbi.nlm.nih.gov/geo/query/acc.cgi?acc=GSE21529 | Publicly available at the NCBI Gene Expression Omnibus (accession no: GSE21529) |

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
