## [Decision Letter]

Thank you for submitting your article "Tbx5-dependent enhancer transcription defines a gene regulatory network for cardiac rhythm" for consideration by *eLife*. Your article has been favorably evaluated by Didier Stainier (Senior Editor) and three reviewers, one of whom is a member of our Board of Reviewing Editors. The following individual involved in review of your submission has agreed to reveal his identity: Philip Grote (Reviewer #2).

The reviewers have discussed the reviews with one another and the Reviewing Editor has drafted this decision to help you prepare a revised submission.

Summary:

This is an excellent paper both conceptually and technically, and contributes to our general understanding of the role of ncRNAs in enhancer function, and more specifically how individual TFs can control a set of TF-dependent ncRNAs that define enhancers functioning in a cell-specific network. The manuscript describes their work to refine the TBX5 dependent gene regulatory network, required to maintain atrial cardiac rhythm. The make use of a diverse mixture of published datasets (TBX5 ChIP, H3K27Ac ChIP, DNaseI HS) and complement them with their own, new datasets (ATAC-seq, RNA-seq). They go on and show that the expression of eRNAs (enhancer RNAs) improves the predictions of functionally relevant atrial TBX5 dependent enhancers. Moreover, they show the functional relevance of some of the newly discovered eRNAs in regulating nearby genes.

The concept that the eRNAs participate in regulating neighboring genes and the TFs such as TBX5 are actively involved in driving eRNAs is not novel. However, some aspects of their work provides further insight (ncRNA-mRNA correlation in expression strength) and the comprehensive identification of functionally relevant TBX5 enhancers is interesting to the cardiovascular community. The novel approach outlined for identifying functional enhancers and associated ncRNAs could see broad utility in systems biology. However, there are some major issues to address as detailed below:

Essential revisions:

1) The manuscript style is very compact. It may help more general readers to expand explanation of methods and graphical approaches. The first part of the paper (text and Figure 1 and Figure 2) is especially difficult to follow and some work (perhaps from the biologist) is needed. We would recommend that the paper be structured around the series of results and clearly relate how one observation leads on to subsequent analyses and results.

2) Main text, third paragraph: regarding hypothesis: presumably the authors mean active enhancers (those functioning in a specific tissue).

3) Main text, third paragraph: please define de novo? Do the authors mean previously unannotated?

4) Main text, fourth paragraph: please clarify uni-directional and bi-directional?

5) Main text, sixth paragraph: how were candidate regulatory elements chosen for transcriptional analysis. Were they chosen from among those with highest number of Tbx5 canonical sites? Please declare.

6) Main text, eighth paragraph:. The story about Sln is confusing. The text might read as though the Sln regulatory element does not contain Tbx5 binding sites as the prelude refers to those that lack Tbx5 binding. But in fact it does (Figure 2). What about the category of elements that do not contain Tbx5 sites. What did they show by Tbx5 ChIPseq and were any analysed by Tbx5 ChIP PCR?

7) Please explain the line scans in Figure 4.

8) Since the experiment was performed using a single knockdown reagent, Figure 4 could benefit from other specificity controls. E.g. was Tbx5 or signalling or metabolic genes (not those highly expressed like GAPDH) unaffected.

9) Main text, fourteenth paragraph and Figure 4. The authors should be very clear about the results of the zebrafish enhancer analysis. As it reads, 67 of 166 embryos showed cardiac expression driven by the RACER-associated enhancer (40%). Mutation of the 7 T-box sites reduced this to 20/84 (23%). This is not a compelling argument for identification of RACER as a Tbx5-dependent element or the function of Tbx5 on this element. While the p value is significant the use of an odds ratio seems rather unusual for this sort of analysis. The assay is poorly quantitative and each injection will lead to a different integration site. Spatial specificity (e.g. atrial) was not scored. I think this should be strengthened or deleted. It would be more compelling to delete the natural enhancer and analyses Ryr2 expression by qRT-PCR.

10) Please explain the line scans in Figure 4. Please correct references to Figure 4 panels in Legends. Check all.

11) Main text, sixteenth paragraph: chromatin enrichment assay. What does this mean? A ncRNA that does not associate with chromatin in the way suggested but, e.g. is a structural component of a TF or splicing complex, may also appear enriched in chromatin. Furthermore, GAPDH seems an inappropriate control. One could use NEAT or another known ncRNA that does not appear to define an enhancer.

12) In the end, the mechanistic link between TF-dependent ncRNA function in transcription of the adjacent genes is not strong. The Pol2 occupancy data is interesting but self-fulfilling. Would knockdown of the RACER change the chromatin landscape around Ryr2 by 4C?

13) It is not clear to the reader why the authors combine H3K27ac and DNAseI HS sites from the whole adult heart with atria specific data (TBX5 ChIP, ATAC, RNA-seq). It should be clearly stated that H3K27Ac and DNaseI HS marks putative, open (active) enhancers in the whole heart, including the atria. Combining ATAC-seq and TBX5 ChIP-seq data are from the HL1 atrial cardiomyocyte cell line and RNA-seq from atrium from cKO mice provide the atria specific data. Please explain in more detail, why you used all the comparison with whole heart data.

14) In the Materials and methods it is mentioned that P300 ChIP-seq data from He et al. was downloaded. Please explain why you didn't use these data for your work? Was it removed? I think could be an interesting addition to see, how the functionally relevant ncRNA regions correlate (or not?) with P300 occupancy.

15) A critical concern is the interpretation of statistical evidence for various enrichment and differential expression analyses. In a number of instances, it appears from the manuscript that significance was determined at a nominal significance of p<0.05 even when a large number of tests were conducted. Was significance determined based on a multiple-testing adjusted p-value?

Throughout the manuscript evidence is often presented in a qualitative manner, making it difficult to judge what weight should be placed on it, and difficult to evaluate the conclusions drawn from it. For example, 'we observed enrichment between…'. It would be helpful to know what level of enrichment and what scale that is on. Also an empirical p-value for said enrichment. Likewise 'we functionally interrogated candidate regulatory elements..' what does functionally interrogate mean? 11/12 have p<0.05, but no correction for that fact that 12 elements tested, so threshold needs to be adjusted. This occurs throughout the manuscript.

I believe that this paper would benefit significantly from using an empirical random-sampling strategy to determine significance of any enrichment. For example, when testing your n TBX5-dependent ncRNAs for enrichment in the 16,000 open regions, randomly sample n regions, determine how many are also candidate Tbx5-dependent enhancers (this value can be called k). Repeat 10,000 times. The position of your observed overlap in the ranked k gives an empirical p-value.

16) In any instance where a p-value is given, you should also provide the test used to estimate it.

---

## [Author Response]

Essential revisions:1) The manuscript style is very compact. It may help more general readers to expand explanation of methods and graphical approaches. The first part of the paper (text and Figure 1 and Figure 2) is especially difficult to follow and some work (perhaps from the biologist) is needed. We would recommend that the paper be structured around the series of results and clearly relate how one observation leads on to subsequent analyses and results.

We agree with the reviewer’s critique and have reorganized and rewritten the manuscript to improve its accessibility. We have made the following changes to the manuscript: 1) We have expanded the methods explanation, Introduction and Discussion as suggested; 2) To clarify the experimental design and results, we have included a bioinformatic pipeline as Figure 1) We have reordered some of the data presentation to be more intuitive; 4) We have including a formal Introduction and Discussion; 5) We have added transitions to each section to clarify the flow of each result to the next; and 6) We have added section breaks to the manuscript.

2) Main text, third paragraph: regarding hypothesis: presumably the authors mean active enhancers (those functioning in a specific tissue).

We appreciate the comment and have clarified this sentence. We have edited the text to read "We hypothesized that transcription factor dependent tissue-specific active enhancers could be identified with this approach".

3) Main text, third paragraph: please define de novo? Do the authors mean previously unannotated?

We appreciate the comment and have clarified this sentence. We now define the term ‘de novo’ as follows: “The vast majority were de novo ncRNAs (>85%), previously annotated transcripts based on intersection with the mouse transcriptome (GENECODE, mm10).”

4) Main text, fourth paragraph: please clarify uni-directional and bi-directional?

We have clarified the definition of uni-directional and bi-directional transcripts. We defined unidirectional ncRNAs as those transcribed from one strand and bi-directional as those transcribed from both strands, and have now clarified this in the text and the Materials and methods sections.

5) Main text, sixth paragraph: how were candidate regulatory elements chosen for transcriptional analysis. Were they chosen from among those with highest number of Tbx5 canonical sites? Please declare.

We previously characterized the requirement for TBX5 in the adult heart for atrial rhythm control and defined several direct TBX5 targets required for cellular and whole animal physiology (Nadadur et al. 2016). Based on this work, we selected TBX5-dependent enhancers near genes we believe may be critical to cardiac physiology. All selected regulatory elements contain at least 1 canonical TBX5 binding motif. We have now explicitly described this approach in the text.

6) Main text, eighth paragraph:. The story about Sln is confusing. The text might read as though the Sln regulatory element does not contain Tbx5 binding sites as the prelude refers to those that lack Tbx5 binding. But in fact it does (Figure 2). What about the category of elements that do not contain Tbx5 sites. What did they show by Tbx5 ChIPseq and were any analysed by Tbx5 ChIP PCR?

We appreciate the comment and have clarified the analysis of the Sln element. This locus is as an example in which the ncRNA approach defined a Tbx5-dependent enhancer that did not show TBX5 binding by ChIP-seq. The primary sequence of the putative enhancer at the Sln locus harbors canonical TBX5 binding motifs. However, genome-wide ChIP-seq (He et al. 2011) did not identify TBX5 localization at this locus. The enhancer generated a *Tbx5*-dependent ncRNA and the enhancer showed TBX5-dependent activity in-vitro. We then performed locus-specific ChIP qPCR and found that TBX5 bound this enhancer in HL-1 cardiomyocytes. This provided an example of a region categorized as a false negative by ChIP-seq, resolved as a true positive Tbx5-bound and Tbx5 driven enhancer by the ncRNA approach and subsequent validation. We have edited the text to clarify this series of experiments and results.

7) Please explain the line scans in Figure 4.

The line scans represent a visual analogue of free intracellular calcium. Time to upstroke, or rate of rise of intracellular calcium, is dependent on the function of the ryanodine receptor encoded by *Ryr2*. We examined the effect of knockdown of the ncRNA *RACER* on calcium handling by HL-1 cells using the line-scan analysis. Specifically, HL-1 cardiomyocytes were transfected with antisense oligonucleotides (ASOs) specific to the *Ryr2* gene, or the *RACER* ncRNA. Time to upstroke, a measure of global Ryr2 activity, was diminished after *Ryr2* knockdown and after *RACER* knockdown. We have modified the text to clarify this series of experiments and results.

8) Since the experiment was performed using a single knockdown reagent, Figure 4 could benefit from other specificity controls. E.g. was Tbx5 or signalling or metabolic genes (not those highly expressed like GAPDH) unaffected.

We agree with this reviewer critique and have added a control knockdown experiment as requested. Specifically, we have performed antisense oligonucleotides against housekeeping gene Hprt (New Figure 4—figure supplement 1). Knockdown of the Hprt mRNA results in no change in *Ryr2* or *RACER* expression compared to untransfected controls.

9) Main text, fourteenth paragraph and Figure 4. The authors should be very clear about the results of the zebrafish enhancer analysis. As it reads, 67 of 166 embryos showed cardiac expression driven by the RACER-associated enhancer (40%). Mutation of the 7 T-box sites reduced this to 20/84 (23%). This is not a compelling argument for identification of RACER as a Tbx5-dependent element or the function of Tbx5 on this element. While the p value is significant the use of an odds ratio seems rather unusual for this sort of analysis. The assay is poorly quantitative and each injection will lead to a different integration site. Spatial specificity (e.g. atrial) was not scored. I think this should be strengthened or deleted. It would be more compelling to delete the natural enhancer and analyses Ryr2 expression by qRT-PCR.

The reviewer is correct to suggest that expression of transgenic reporters to test enhancer activity in individual microinjected zebrafish embryos (F0) is mosaic and impacted by insertion site. As an additional experiment, we have now selected independent stable lines (F1) which demonstrate cardiac expression domains overlapping with Myosin Heavy Chain expression (Figure 4, Figure 4—figure supplement 2), providing further evidence that RACER can direct gene expression in the zebrafish myocardium.

Close comparison of the activities of a wild-type enhancer and its mutated form can be challenging even using stable lines. Knockout of an endogenous candidate enhancer is a good experiment, although this would require several months to complete. Furthermore, a negative result in these knockouts would not exclude that a sequence has enhancer activity; for example, there may be compensation by shadow enhancers. Here, using microinjected embryos, we acquire high numbers necessary to provide an estimate of enhancer activity. We now include a negative control to make this point more clearly – injection of the minimal promoter driving GFP on its own only yields detectable fluorescence in only 5 of 95 embryos. This suggests that insertional effects account for little of the observed cardiac EGFP, and that removal of Tbx5 binding sites reduces efficiency of the reporter in a statistically significant manner.

10) Please explain the line scans in Figure 4. Please correct references to Figure 4 panels in Legends. Check all.

We have clarified the line scans as described above. Please see response to comment #7.

11) Main text, sixteenth paragraph: chromatin enrichment assay. What does this mean? A ncRNA that does not associate with chromatin in the way suggested but, e.g. is a structural component of a TF or splicing complex, may also appear enriched in chromatin. Furthermore, GAPDH seems an inappropriate control. One could use NEAT or another known ncRNA that does not appear to define an enhancer.

We appreciate the comment and have clarified the chromatin enrichment assay in the text. Specifically, nuclei from HL-1 cardiomyocytes were isolated and fractionated into soluble and chromatin enriched fractions as previously described (Werner and Ruthenburg, 2015; Werner et al. 2017). Chromatin enriched is defined as a transcript that was found at a statistically greater degree in the chromatin vs. soluble nuclear fractions (see Materials and methods for details). While we agree that the precise mechanism of attachment that gives rise to chromatin enrichment is not clear from the experiment presented, the observation that RACER and several other TBX5-dependent noncoding RNA are tightly chromatin associated is evident.

We appreciate the reviewer suggestion for additional controls, including chromatin-enriched ncRNA. It is not immediately clear to us why a coding transcript as a negative control is inappropriate. GAPDH displays high steady state transcription so some portion of it will be associated with chromatin, yet the bulk of the nuclear pool in four different cell types we have worked with is strongly nucleoplasmic (this manuscript, Werner and Ruthenburg, 2015; Werner et al. 2017). We have now added a positive control, XIST, a noncoding RNA that is tightly associated with chromatin, but not merely at its site of production (Engreitz et al., Science, 2013). We have now found that Xist demonstrates chromatin enrichment in HL-1 cardiomyocytes, and included this in the revised manuscript (Figure 4). We have performed sequencing on these two sub-nuclear fractions of RNA in several other cell types and found these to be very effective negative and positive controls for chromatin association (Werner and Ruthenburg. 2015; Werner et al. 2017). It should be noted that neither of these RNAs defines an enhancer.

12) In the end, the mechanistic link between TF-dependent ncRNA function in transcription of the adjacent genes is not strong. The Pol2 occupancy data is interesting but self-fulfilling. Would knockdown of the RACER change the chromatin landscape around Ryr2 by 4C?

The precise mechanism by which enhancer-generated ncRNAs participate in transcriptional regulation is a controversial topic, and in the context of TF-dependent ncRNAs is worthy of its own manuscript. The question of whether RACER participates in chromatin architecture around Ryr2 is an interesting one. It is fair to say that the architectural model presented suggested by the reviewer is a reasonable inference from the available data: 1) *RACER* is highly chromatin immobilized; 2) targeting RACER at the RNA level with antisense oligonucleotides reveals *Ryr2* transcriptional deficits, indicating that the RNA molecule of *RACER* plays a critical role in potentiating *Ryr* transcription; 3) Other chromatin enriched RNAs are largely immobilized by the act of transcription (Werner and Ruthenburg, 2015; Schlackow et al., Mol Cell 2017); 4) Other similar chromatin immobilized lncRNA display similar architectural properties [Werner et al. (2017); Lai et al. Nature, 494: 497–501 (2013).; Lai et al. Nature, 525: 399–403 (2015); Yang et al. Nature, 500: 598–602 (2013); Li et al. Nature Biotech 35: 940–950 (2017)]. However, 4C results, regardless of their outcome in this case, would not alter the major conclusions of our manuscript. We will pursue such studies in the context of ongoing investigations into the mechanism of action of enhancer-generated TF-dependent ncRNAs.

13) It is not clear to the reader why the authors combine H3K27ac and DNAseI HS sites from the whole adult heart with atria specific data (TBX5 ChIP, ATAC, RNA-seq). It should be clearly stated that H3K27Ac and DNaseI HS marks putative, open (active) enhancers in the whole heart, including the atria. Combining ATAC-seq and TBX5 ChIP-seq data are from the HL1 atrial cardiomyocyte cell line and RNA-seq from atrium from cKO mice provide the atria specific data. Please explain in more detail, why you used all the comparison with whole heart data.

We appreciate the comment and acknowledge that where there are distinctions in the tissue source for the genomic dataset they should be clearly identified. We have rectified this oversight in our revised manuscript. We mined the existing literature for published genomic datasets to assist in identifying atrial regulatory elements. When possible, we utilized atrial specific datasets. However, there are no published atrial specific datasets for some enhancer marks, such as H3K27Ac or DNaseI. Given the concordance between HL1, whole heart, and in vivo atrial data when available (Figure 1), we independently mined these datasets in conjunction with atrial-specific Tbx5-dependant ncRNA-seq to identify candidate TBX5-dependent elements.

14) In the Materials and methods it is mentioned that P300 ChIP-seq data from He et al. was downloaded. Please explain why you didn't use these data for your work? Was it removed? I think could be an interesting addition to see, how the functionally relevant ncRNA regions correlate (or not?) with P300 occupancy.

Utilizing P300 occupancy in conjunction with the other enhancer marks to identify regulatory elements is an excellent suggestion. The P300 ChIP-seq was downloaded as part of the larger GSE21529 dataset from which we utilized TBX5 ChIP (He et al. 2011). However, we found that the cardiac P300 ChIP-seq dataset was too stringent for our purposes. The very small number of peaks called (n=1491) caused the overlap with Tbx5-dependent ncRNAs to be too small for meaningful analysis.

15) A critical concern is the interpretation of statistical evidence for various enrichment and differential expression analyses. In a number of instances, it appears from the manuscript that significance was determined at a nominal significance of p<0.05 even when a large number of tests were conducted. Was significance determined based on a multiple-testing adjusted p-value?

We agree wholeheartedly with this critique, applied stringent standards for multiple testing corrections in our analysis, and acknowledge that we did not make our efforts explicit in the original submission. For all tests of significance using genomic datasets (RNASeq, ChIPSeq, etc) we utilized FDR or corrected p-values in accordance with ENCODE published standards (Consortium, E.P. An integrated encyclopedia of DNA elements in the human genome. Nature 489, 57-74, 2012). We have specified "corrected p-value" or "FDR" in the text wherever appropriate, and have stated this explicitly in the Materials and methods statistics discussion.

Throughout the manuscript evidence is often presented in a qualitative manner, making it difficult to judge what weight should be placed on it, and difficult to evaluate the conclusions drawn from it. For example, 'we observed enrichment between…'. It would be helpful to know what level of enrichment and what scale that is on. Also an empirical p-value for said enrichment.

We agree that quantitation and statistical testing of the comparisons is critical to interpreting the data. In the interest of allowing easier reading, we specified the numerical p-values in the figures, figure legends, or supplementary data explicitly. As suggested, we have now specified these values in the text and estimated the empirical p-values for the enrichment between open chromatin and Tbx5-dependent ncRNAs (Figure 1—figure supplement 4), and also between the Tbx5-dependent ncRNAs and the adjacent Tbx5-dependent genes (Figure 1—figure supplement 3). Please see our fourth response to #15 below.

Likewise 'we functionally interrogated candidate regulatory elements..' what does functionally interrogate mean? 11/12 have p<0.05, but no correction for that fact that 12 elements tested, so threshold needs to be adjusted. This occurs throughout the manuscript.

We cloned regulatory elements upstream of a luciferase reporter vector and tested them for activity in HL-1 cardiomyocytes, as previously described (Nadadur et al. 2016). For the p-values stated on the figure, each regulatory element was tested for significant activity against its corresponding T-box mutant element. As such, each experiment is an independent comparison using DNA elements unique to the comparison, multiple testing correction was not applied.

I believe that this paper would benefit significantly from using an empirical random-sampling strategy to determine significance of any enrichment. For example, when testing your n TBX5-dependent ncRNAs for enrichment in the 16,000 open regions, randomly sample n regions, determine how many are also candidate Tbx5-dependent enhancers (this value can be called k). Repeat 10,000 times. The position of your observed overlap in the ranked k gives an empirical p-value.

We thank the reviewer for this insightful suggestion. We have performed and added an empirical random-sampling test for the enrichment between Tbx5-dependent ncRNAs and chromatin open regions and for the enrichment between Tbx5-dependent ncRNAs and Tbx5-dependent genes.

We have provided a description of this simulation in the supplementary methods, as follows:

“Simulation study on open chromatin regions that were TF-

To test the 3067 TBX5-dependent ncRNAs for enrichment in chromatin open regions, we randomly sampled 3067 regions from the background of total identified chromatin open loci from atrial HL-1 cells (16,000). […] This provided an empirical p-value of P<0.0001 (Equation 2).”

We revised our manuscript describing the enrichment between Tbx5-dependent ncRNAs and chromatin open regions to read “From 16,000 open regions, 145 local Tbx5-dependent ncRNAs marked 152 candidate Tbx5-dependent enhancers (empirical P<0.001, Figure 1—figure supplement 4)” and for the enrichment between Tbx5-dependent ncRNAs and Tbx5-dependent genes to read “*Tbx5*-dependent ncRNAs were local to down-regulatedand up-regulated *Tbx5*-dependent coding-genes, respectively (Figure 1, FA, Figure 1—figure supplement 3, empirical P< 0.005).”

16) In any instance where a p-value is given, you should also provide the test used to estimate it.

We thank the reviewer for noting this oversight. We have now included the test used to analyze for significance in each case on the text.